# Physical Therapy Interventions in Patients with Anorexia Nervosa: A Systematic Review

**DOI:** 10.3390/ijerph192113921

**Published:** 2022-10-26

**Authors:** Emilio Jose Minano-Garrido, Daniel Catalan-Matamoros, Antonia Gómez-Conesa

**Affiliations:** 1International School of Doctoral Studies, University of Murcia, 30100 Murcia, Spain; 2Department of Nursing, Physiotherapy and Medicine, and Health Research Centre, University of Almería, 04120 Almería, Spain; 3Institute for Culture and Technology, Madrid University Carlos III, 28903 Madrid, Spain; 4Research Group Research Methods and Evaluation in Social Sciences, Mare Nostrum Campus of International Excellence, University of Murcia, 30100 Murcia, Spain

**Keywords:** anorexia nervosa, supervised exercise, physical activity, physical therapy modalities, women’s health

## Abstract

Objective: Assess the effectiveness of physical therapy, including supervised physical exercise for body mass index (BMI) restoration; improving muscle strength and the psychological, behavioural, cognitive symptoms and quality of life in patients with anorexia nervosa (AN). Methods: A Systematic Review (SR) was conducted in the following scientific databases: Medline, PubMed, PEDro, PsychInfo, Cochrane Library plus, Nursing and Allied Health database, Scopus and Web of Science databases, from inception until November 2021. An assessment of the risk of bias and the certainty of evidence across studies was conducted. Articles were eligible if they followed randomized and non-randomized control trial designs with treatments based on physical therapy or exercise or physical activity in AN patients. Results: 496 records were screened, and after eligibility assessment, 6 studies from 8 articles were finally analysed. The studies, involving 176 AN patient (85.02% of patients), reported improvements in muscle strength, eating behaviour, eating attitude, mood and quality of life. Three studies included nutritional co-interventions and four studies included psychological therapy. None of the studies reported adverse effects. Conclusions: In two of the RCTs included in this SR, strength training and high intensity resistance improved the muscle strength of patients with AN. In addition, in two RCTs, improvements were observed in patients’ attitudes towards their bodies after basic body awareness therapy or after full body massage and instruction to relax. In addition, quality of life improved in two studies, with stretching, isometrics, endurance cardiovascular and muscular exercising.

## 1. Introduction

Anorexia nervosa (AN) is a serious, complex psychiatric disorder with potentially severe somatic consequences, and along with bulimia nervosa (BN) is one of the main eating disorders (ED), which can even be life-threatening for patients [1,2]. The prevalence of AN varies between 0.9% and 3% in the general population [3], while incidence is ten times higher in women [4]. Moreover, it is the psychiatric disorder with the highest mortality rate, with a mean 5.2 times higher than that of the general population of the same age [5,6]; it may even reach 10% in hospitalised patients [7], and over 12% in patients who suffer AN for over 10 years [7,8].

AN is characterized by a restriction of energy intake relative to needs, leading to significantly low body weight, along with altered self-perception of one’s own weight or build, and intense fear of gaining weight or becoming fat, even with a significantly low weight [6]. This leads to self-induced inanition and excessive physical exercising in 80% of patients suffering from AN [9,10]. This excessive exercising may have an even higher impact than anxiety or depression, and the negative effects of maintaining the pathological behaviour and the disorder [11], could also have a negative impact on the quality of life of severe ED patients [10,12]. In addition, hyperactivity during the course of AN is an indicator of poor clinical outcome [13,14]. Furthermore, as is the case in other ED, comorbidity with other mental problems such as depression, anxiety, bipolar disorder and substance abuse is common in AN patients [15]. EDs are one of the hardest mental disorders to address and treat, and the cost is even higher than in schizophrenia [16].

Secondary somatic complications to undernutrition, severe weight loss and low levels of micronutrients, in addition to heart problems (bradycardia) [17], hormonal problems (amenorrhea), intestinal problems or bone demineralisation, have also been studied [18,19]. Nevertheless, the loss of autonomy and reduction in muscle strength are not sufficiently described in the scientific literature [20]. To date, only a few clinical observations in small groups have been published [9,21].

The physical deterioration of people with anorexia nervosa suffering from undernutrition can also lead to lower respiratory capacity, thereby increasing the risk of respiratory infection. Consequently, the loss of autonomy and reduced respiratory capacity can cause a worsening of the vital prognostic in these patients, particularly in patients with severe undernutrition [22].

A large number of therapy interventions have been tested on AN, such as cognitive behavioural therapy, psychosocial therapy, medication, nutritional intervention, etc. Physiotherapy treatments have been under development since the 1980s [9,23].

As supervised physical therapy might increase weight and improve the overall condition in AN patients through therapeutic aerobic exercising, massage, basic body awareness therapy and yoga, these interventions could reduce the eating disorder in patients with AN and BN [9]. In addition, aerobic exercise, yoga and basic body awareness therapy might also improve the mental and physical quality of life in patients with eating disorders [9]. Current knowledge suggests that maintaining some kind of physical activity during refeeding of AN patients should be safe and beneficial for the restoration of body composition and bone mineral density, as well as mood and anxiety [13]. Therefore, this study aimed to (1) provide an updated, systematic review of the evidence related to physiotherapy modalities, including supervised physical exercise for weight recovery, improving muscle strength and quality of life in AN patients, and (2) verify the safety of this treatment to ensure no further weight loss takes place.

## 2. Methodology

A systematic review (SR) was conducted independently by the three authors, following the recommendations proposed by the Preferred Reporting Items for Systematic Review and Meta-Analysis [24]. This SR was prospectively registered in the International Prospective Register of Systematic Reviews (PROSPERO) database under number (CRD42020176679).

### 2.1. Inclusion Criteria

Studies were included in accordance with the following criteria: (1) Randomized controlled trials (RCTs) and controlled clinical trials (CCTs); (2) All the trials had to have at least five analysed subjects in each of the groups at the end of the trial (including the control group); (3) Patients diagnosed with AN following the diagnostic criteria established in the Diagnostic and Statistical Manual of Mental Disorders (DSM) by the American Psychiatric Association (APA) [6] or the International Classification of Diseases (ICD-10) by the World Health Organization (WHO) [25]; (4) The trials could include patients with other eating disorders, although at least 70% of the patients should have been diagnosed with AN; (5) Studies were included if participants were adolescents or adults; (6) Studies conducting an analysis of the effects of physiotherapy modalities. Physiotherapy modalities are defined as: *“Therapeutic modalities frequently used in the physical therapy specialty to promote, maintain, or restore the physical and physiological well-being of an individual”* [26], including exercising and physical activity; (7) Treatment could be administered by a physical therapist or a qualified healthcare provider; (8) Studies with hospitalised and non-hospitalised patients with medical monitoring during treatment; (9) Studies assessing the effects of treatment on physiological and muscular aspects and the psychological, behavioural or cognitive symptoms, or quality of life; (10) No language restrictions were enforced. Studies were excluded if original data were not used (e.g., literature or systematic reviews).

### 2.2. Literature Search and Study Selection

The literature search covered the period from the start of the search until November 2021, in the Medline, PubMed, PEDro, PsychInfo, Cochrane Library plus, Nursing and Allied Health database, Scopus and Web of Science databases. The search strategy included the target condition and target intervention terms. The electronic database search strategy is included in the Appendix A.

The number of retrieved reports, the number of included and excluded studies and the reasons for exclusion are presented in Figure 1 [27].

### 2.3. Data Extraction and Risk-of-Bias Assessment

Data extraction was carried out independently by the three authors. In the event of disagreement, the three authors rechecked the original article and following up with a discussion in order to reach a consensus.

To evaluate the results of the aforementioned studies, the differences between pre-treatment and post-treatment and the differences between groups (experimental and control) were analysed. The analysed variables were as follows: (1) Physiological and muscle results: body mass index (BMI), body fat, muscle strength and skeletal muscle mass; (2) Psychological results: depression, anxiety, negative effect; (3) Behavioural results: eating, impulse control; (4) Cognitive results: emotions, moods, beliefs, attitudes; (5) Quality of life results.

The three authors independently assessed the risk of bias (RoB) for each study using the Cochrane Collaboration’s tool for assessing RoB [28]. Inconsistencies in judgements were resolved through discussion or by involving the three authors.

### 2.4. Certainty of Evidence

The certainty of evidence across studies was assessed using GRADE [29], and was summarized in four categories: “high”, “moderate”, “low”, “very low”; discrepancies were resolved through consensus by the three authors. The GRADE approach takes the study limitations (RoB), inconsistency of results, indirectness of evidence, imprecision and publication bias into consideration.

## 3. Results

### 3.1. Search Results

After discarding articles that did not meet the inclusion criteria, 496 records were screened. After screening, 34 full-text articles were assessed for eligibility. Finally, 6 studies from 8 articles were included in this SR (Figure 1) [27].

### 3.2. Study Characteristics

Six studies (all of them RCTs), including 207 patients, all of them with eating disorders, of which 176 suffered from AN, were included in this SR [30,31,32,33,34,35,36,37]. The main characteristics of these studies are summarized in Table 1. The study publication dates are between 2000 [30] and 2018 [37], and the sample sizes range from 20 [30] to 78 patients [37].

Study duration varied from five weeks in one of the studies [31], one month and 19 days in another study [33], two months in another study [34,35,36], three months in two other studies [30,32], and 8.5 months in the longest study [37]. The mean duration of the studies included in the SR is 13 weeks.

The six studies administered a total of 146 treatment sessions, with a mean of 1.87 sessions per week and a frequency of two sessions per week in four of the studies [31,32,33,37] and three sessions per week in the other two studies [30,34,35,36]. The average session duration was 57 minutes.

Of the 207 patients, 176 (85.02%) had been diagnosed with AN, 18 (8.69%) with bulimia nervosa (BN), and 13 (6.28%) with an eating disorder not otherwise specified (EDNOS). One hundred and twenty-five patients (60.38% of the total) in three of the studies were aged over 18 years [31,33,37], 66 patients (31.88%) in two of the studies were adolescents aged between 12 and 16 [32,34,35,36] and 16 patients (7.73%) were aged over 17 [30].

Of the 207 patients included in the six studies, 198 were females (95.65%), whereas the remaining 9 males were distributed over four studies [31,32,33,37]. Of the patients, 90.82% (188) were not hospitalised [30,31,32,33,34,35,36,37] and one study with 19 patients included hospitalised and non-hospitalised patients, although the number of each is not specified [31].

Three studies include information on patients’ nutritional status or nutritional needs: instruction on nutrition and principles of physiology and metabolism [30]; daily calorie intake in the range of 2000–2500 kcal/day depending on patients’ weight [31]; energy intake control with 55% carbohydrate, 30% protein, and 15% fat, and also additional increased calorie intake (~150 kcal) in the experimental group because of the energy expenditure required during the strength training session [34,35,36].

Likewise, four studies include some modality of psychological therapy as co-intervention [31,32,33,34,35,36].

### 3.3. Physiological and Muscular Effects

At the start, the mean BMI of the patients in the six studies was 18.10 in the treatment group and 17.59 in the control group. By study, the lowest BMI was 16.47 [37], and the highest was 20.26 [30]. In relation to the BMI of the patients in the primary studies included in our SR, Thien et al. [30], Catalan-Matamoros et al. [33], and Hart et al. [31] do not specify a BMI in their inclusion criteria. In the studies of del Valle et al. [32,34,35,36], they required a BMI higher than 14, and in the study of Hay et al. [37], the patients could have a BMI range between 14 and 18.5; therefore, the thinness would be between mild and severe, according to the WHO.

As a result of treatment, the BMI increased significantly in one study [37], and the Arm Muscle Area (AMA) and Mid-thigh Muscle Area (MTMA) increased in another study [34,35,36]. In regard to strength, this increased after three months of training [32], and muscle strength increased after two months of endurance training, bench press, leg press and side rowing [34,35,36].

### 3.4. Psychological and Cognitive Effects

After seven weeks with Basic Body Awareness Therapy (BBAT) and five weeks with full body massage and instruction to relax, improvements were observed in patients’ attitudes towards their bodies [31,33]. Anxiety was also reduced, and mood improved with full body massage and instruction to relax [31].

### 3.5. Behavioural Effects

After 8.5 months of physical exercising treatment and compulsive exercising control, a reduction in the addiction to exercise was achieved [37]. Regarding eating behaviour [31,33], significant improvements were achieved after five weeks of full body massage and instruction to relax [31], and after seven weeks with BBAT [33].

### 3.6. Effects on Quality of Life

Two studies reported improvements to quality of life. Both the treatment over 3 months with stretching, isometric, endurance cardiovascular and muscular exercising [30], and treatment consisting of adding exercising and compulsive exercising control over 8.5 months to cognitive behavioural therapy (CBT) [37], reported significant differences in favour of the treated group.

Table 1 shows the results of the studies included in this SR.

### 3.7. Risk of Bias and Certainty of Evidence

The RoB assessment for each individual study is presented in Table 2. The studies show a low risk of random sequence generation (100%), allocation concealment (83%), incomplete outcome data (83%), selective outcome reporting (83%) and other bias (83%). On the other hand, they also show a high risk of the blinding of participants and personnel (17%) and the blinding of outcome assessment (17%).

A summary of the rate of certainty of evidence is provided in Table 3. Of the ten results whose effects were evaluated, six were considered to be of moderate certainty and four of low certainty, and there was a decreasing range from 5/9 to 2/9 (likewise considered from moderate to limited importance).

## 4. Discussion

This SR provides a comprehensive appraisal of the current evidence concerning the role of physical therapy modalities, including supervised physical exercise, in the treatment of AN. In total, six RCTs, including 207 patients with eating disorders, of which 85.02 % suffered from AN, were included in this SR. This is the first SR on the effects of physiotherapy treatment on eating disorders that includes such a wide proportion of patients with AN. In previous SR on physical therapy interventions in AN, the patients diagnosed with AN included in the study accounted for approximately 40% of a sample of 786 patients [9,38,39].

Across the six eligible studies, from the eight articles, patients with AN had a mean BMI of 18.10 (experimental group) and 17.59 (control group), in both cases close to a BMI of 17.5, which, in adults, is one of the common physical characteristics used to diagnose AN. In this sense, four studies evaluated the effects of treatment on BMI [30,32,34,35,36,37], with no noteworthy results because no improvements were observed, or no differences between the treatment and control groups were observed, although improvements were observed in one study, but this was after 8.5 months of treatment [37].

In previous SR, the results of physiotherapy on BMI are similar, finding no differences between the intervention group and the control group [38,39], or indicating an increase in BMI in two of the reviewed studies [9].

Regarding muscle strength, this was assessed in two studies [32,34,35,36], and in both cases an increase in muscle strength was observed through the 6RM test after 12 weeks of strength training, with and without weights [32]; and Bench Press and Leg Press tests after 8 weeks of high-intensity strength training [34,35,36]. Previous SR point out increases in muscle strength in two studies [9] and three studies [39], and observed improvements in muscle strength and cardiovascular endurance [38].

With respect to the psychological effects, after seven weeks of BBAT training and five weeks of massage treatment, improvements were observed in patients’ attitudes towards their bodies [31,33], and after five weeks of massage treatment, anxiety was reduced, and mood was improved [31]. Previously, Vancampfort et al. [9] observed a reduction in depression symptoms through physical therapy treatment in AN and bulimia nervosa patients. Likewise, Ng et al. [38] and Moola et al. [40] reported improvements to the psychological wellbeing of patients with AN who had been treated with adapted physical exercising.

Regarding behavioural effects, in the studies where eating behaviour was assessed [31,33,37], significant improvements were achieved in two of them through the use of BBAT [33] and massage [31]. Similar results were reported in other SR in respect of a reduction in eating behaviour symptoms [9,38,39]. However, in relation to quality of life, findings are not so aligned. Quality of life was assessed in four of the six studies [30,32,33,37], and contradictory results were reported concerning the improvement [30] and lack of differences between the groups [32], and even in regard to both results in the same study depending on the administered test [37]. Likewise, previous SR were not conclusive on this subject. On the one hand, they reported improvements to the quality of life of patients with AN who were given physiotherapy or adapted physical exercise [9,40], while on the other hand no improvements were observed [38,39].

No RCT included in this SR referred to adverse effects or complications caused through participation by AN patients in the clinical interventions. Along the same lines, Vancampfort et al. did not report any complications due to physical therapy in eating disorders [9].

In general terms, we can see that the studies on physiotherapy treatment in women with AN, including adapted physical exercise, have mainly been carried out over the last 20 years, and the increase in the sample sizes is also noteworthy. Both aspects are indicative of the growing interest in physiotherapy treatment on mental health, and more specifically eating disorders and AN. In regard to the RoB assessment, the risk is low with the exception of blinding of participants and personnel and blinding of outcome assessment. One explanation regarding the high risk of the blinding of participants during assessment in the studies could be due to the fact that it is not possible to truly blind patients to treatment allocation in exercise training. In this respect, one of the studies suggests that after asking 50% of the participations about blinding, 64% correctly identified their treatment group [37]. As for the certainty of evidence, there were six RCTs that informed about the effect of the treatment on AN. In this SR, five analysed results were considered of importance and of moderate certainty: physiological and muscular effects, measured through BMI or muscle strength; cognitive mood effects (assessed using Profile of Mood States) and attitudes (measured using the Body Attitude Test and Eating Attitude Test-40) and the effects on eating behaviour (measured using Eating Disorders Inventory, Motivation to change 20-item AN Stages of Change Questionnaire and Eating Disorder Examination Questionnaire).

### 4.1. Implications for Rehabilitation

AN is a serious and complex psychiatric disorder, with severe somatic consequences, poor prognostics, increased mortality, decreased quality of life and, in the most serious cases, reduced autonomy and physical capacity.

Supervised physical therapy might increase weight or BMI, and improve muscle strength and cardiovascular endurance, and the overall condition in AN women. In relation to quality of life, there is still controversy as previous SR were not conclusive on this subject.

In previous systematic reviews carried out regarding eating disorders, less than 40% of participants had been diagnosed with AN. This is the only systematic review carried out with more than 85% of participants diagnosed with AN.

### 4.2. Strengths and Limits

This systematic review covers underlying physiotherapy modality characteristics, including supervised physical exercise, that may affect weight recovery, muscle strength and quality of life in women with AN. Some of this review’s strengths are based on its search strategy protocol, including an independent search by three researchers with expertise, and searches in a variety of complementary electronic databases. In addition, the results, whose effects were evaluated, have been reported by GRADE. Finally, in comparison with previous ED review studies, this SR includes a greater number (85.02%) of participants that were diagnosed with AN.

There are several limitations to this study, mainly the integration of a small number of studies with heterogeneous interventions in terms of content and duration and to the evaluation times. Furthermore, the mean BMIs are 18.10 and 17.59 in the treatment and control groups, respectively, and therefore generalisation of findings for patients with lower BMI is limited, particularly where low BMI is concerned, owing to the serious physical complications involved with clearly low body weight, loss of autonomy or respiratory complications [13,22]. 

## 5. Conclusions

This review provides support for strength training and high intensity resistance for AN women in order to improve muscle strength, as well as BBAT and full body massage and instruction to relax, thus giving support to improve eating behaviour. In addition, BBAT enhances their attitudes to their bodies, while full body massage and instruction to relax improve mood and reduce anxiety. Stretching, isometric, endurance cardiovascular and muscular exercising, as well as adding exercising and compulsive exercising control to CBT, improve quality of life. Furthermore, none of the studies reported any adverse effects caused by the experimental treatment.

However, these findings should be interpreted with caution because of the quality of the evidence in the heterogeneity we found. In order to validate these results, we recommend large higher-quality RCTs, including comparisons of more physical therapy modalities.

## Figures and Tables

**Figure 1 ijerph-19-13921-f001:**
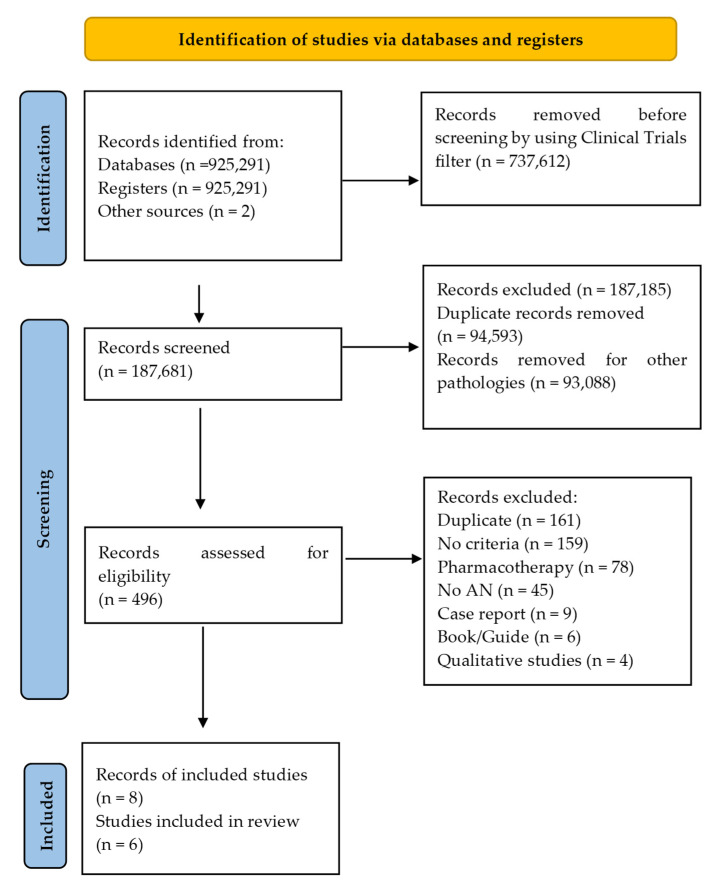
Study flow diagram [27].

**Table 1 ijerph-19-13921-t001:** Summary of characteristics of included studies.

Scheme.	Participants N and Diagnosis; *n* Females/Males; (% Female); Mean Age (Years Treatment/Control)	Treatment (Numbers) [Control (Numbers)]	Duration; Frequency (Sessions/wk); min/Session; [Training: Individual, Group, or Unspecified]	Co-Interventions	Outcome Measures	Results Treatment vs. Control (from Baseline to Post-Test)	Adverse Effects
Thien et al., 2000 [30]	16 AN; 15/ 1; 93.75%; 29/36.1	Stretching, isometrics, cardiovascular, resistive stretching, increase weekly (*n* = 8) [Standard Care and Limit exercise (*n* = 8)]	3 months; 3/wk [Training: unspecified]	Standard Care: Psychiatric follow-up	BMI BF QOL: SF-36	≠BMI (from 20.26 ± 1.8 vs. 17.2 ± 1.6 at baseline; to increase 1.0 ± 1.3 vs. increase 0.8 ± 1.1 at 3 months; *p* = 0.37) ≠BF (from 21.0 ± 2.9 vs. 16.7 ± 4.9 at baseline; to increase 0.9 ± 2.1 vs. increase 0.5 ± 2.6 at 3 months; *p* = 0.39) ↑ SF-36 (from 58.8 ± 13.9 vs. 53.3 ± 14.5 at baseline; to increase 6.6 ± 7.0 vs. increase −12.0 ± 25.5 at 3 months; *p* = 0.07)	Not reported
Hart et al., 2001 [31]	19 AN; 19/0; 100%; 24.9/26.3	Full body massage Instruction to relax and discourage her from talking (*n* = 10) [Standard Care *n* = 9]	5 weeks; 2/wk; 30 min. [Training: unspecified]	Standard Care: Psychiatric and nutritional follow-up Engaged other activities, such as movement therapies	Anxiety: STAI Mood States: POMS Eating Disorders: EDI	↑ STAI (from 52.3/40.1 vs. 50.2/46.5 at baseline; to 47.3/37.0 vs. 48.8/46.1 at 5 wk; *p* < 0.01) ↑ POMS (from 39.5/28.5 vs. 32.8/31.7 at baseline; to 30.0/20.2 vs. 29.1/25.0 at 5 wk; *p* < 0.05) ↑ EDI (from 88.8 vs. 83.1 at baseline; to 66.0 vs. 80.5 at 5 wk; *p* < 0.05)	Not reported
Fernandez-Del-Valle et al., 2010 [32]	22 AN; 20/2; 90.90%; 14.7 ± 0.6/14.2 ± 1.2	Strength training, Isometric contraction with their body weight or barbells of 1–3 kg (*n* = 11) [Standard Care Control and maintain physical activity level *n* = 11]	12 weeks; 2/wk; 60–70 min. [Training: individual]	Standard Care: Psychotherapy Dietary counseling	BMI BF Muscle strength: 6RM	≠BMI (from 18.7 ± 1.7 vs. 18.2 ± 1.5 at baseline; to 18.2 ± 2.2 vs. 18.3 ± 1.6 at 12 wk; *p* = 0.543) ≠BF (from 14.8 ± 2.9 vs. 13.8 ± 2.6 at baseline; to 13.7 ± 3.3 vs. 13.9 ± 2.4 at 12 wk; *p* = 0.247) ↑ 6RM (from 45.4 ± 5.7 vs. 46.0 ± 11.8 at baseline; to 59.0 ± 13.7 vs. 46.9 ± 11.3 at 12 wk; *p* = 0.009)	Program well tolerated
Catalan-Matamoros et al., 2011 [33]	8 AN; 10 BN; 3 atypical AN; 1 atypical BN 6 non-detailed diagnosis (28) 26/2; 92.85%; 29.5/25.2	Basic Body Awareness Therapy (*n* = 14) [Standard Care Control *n* = 8]	7 weeks; 1/wk; 1 h. (during 2 wk); 2/wk; 1.5 h. [Total 12 sessions/patient] [Training: individual and group]	Standard care: Psychotherapy and Psychiatric follow-up	Eating Disorder: EDI Body Attitude: BAT Eating Attitude: EAT-40	↑ EDI (33.6 ± 32.0EG vs. 7.3 ± 13.7CG at 7 wk; *p* = 0.015) ↑ BAT (24.7 ± 26.4 EG vs. −8.3 ± 28.4CG at 7 wk; *p* = 0.012) ↑ EAT-40 (14.8 ± 21.8 EG vs. 1.2 ± 4.1 CG at 7 wk; *p* = 0.039)	Not reported
Fernandez-Del-Valle et al., 2014; 2015; 2016 [34,35,36]	44 AN-R; 44/0; 100%; 12.61/13.0	High intensity resistance program: Isometrics and stretching (*n* = 18) [Standard Care Control *n* = 18]	8 weeks; 3 /wk; 50–60 min. [follow-up 4 wk] [Training: individual]	Standard Care: Psychological therapy and control calorie intake	BMI BF Muscle strength: Bench press; Leg press; Lateral row Muscular areas: AMA; MTMA Relative strength: Strength to BW; Strength to SMM	≠BMI (from 17.28 ± 2.55 vs. 18.12 ± 2.11 at baseline; to 17.82 ± 2.50 vs. 18.50 ± 2.10 at 8 wk; p=0.242) ≠BF (from 25% vs. 26% at baseline; to 24% vs. 27% at 8 wk; *p* = 0.075) ↑ Bench press IG 41% (mean dif. –1.65) at 8 wk; *p* < 0.001 ↑ Leg press IG 52% (mean dif. 3.80) at 8 wk; *p* < 0.001 ↑ Lateral row IG 37% (mean dif. –2.27) at 8 wk; *p* < 0.001 ↑ AMA (from 21.3 ± 5.2 vs. 25.5 ± 5.6 at baseline to 23.3 ± 5.7 vs. 24.1 ± 5.1 at 8 wk; *p* = 0.030 ↑ MTMA (from 122.99 ± 19.4 vs. 137.63 ± 20.3 at baseline to 128.00 ± 20.8 vs. 133.74 ± 17.4 at 8 wk; *p* = 0.061) Strength to BW at 8 wk: ↑ Bench press 32.97% (27.14%–49.29%); *p* < 0.001 ↑ Leg press 49.59% (32.93%–66.24%); *p* < 0.001 ↑ Lateral row 38.22% (22.53%–43.40%); *p* < 0.001 Strength to SMM at 8 wk: ↑ Bench press 37.11% (28.20%–46.01%); *p* < 0.001 ↑ Leg press 49.11% (31.79%–66.43%); *p* < 0.001 ↑ Lateral row 32.56% (21.39%–43.72%); *p* < 0.001 Strength to BW follow-up 4 wk: ↑ Bench press (mean dif. –0.23 vs. −0.01); *p* < 0.001 ↑ Lateral row (mean dif. –0.15 vs. −8.8); *p* = 0.014 Strength to SMM follow-up 4 wk: ↑ Bench press (mean dif. –0.68 vs. −0.04); *p* < 0.001 ↑ Leg press (mean dif. −0.79 vs. –0.13); *p* = 0.039 ↑ Lateral row (mean dif. –0.44 vs. −0.27); *p* = 0.005	No adverse effects or health problems
Hay et al., 2018 [37]	78 AN; 74/4; 94.87%; 26.1/28.6	Compulsive Exercise Activity Therapy (LEAP) (*n* = 39) [CBT-AN Control *n* = 39]	8.5 months; 2/wk (during 4 wk); 1/wk; 50 min. [Intervention 8 LEAP sessions and 26 CBT-AN sessions; Control 34 CBT-AN sessions] [Training: individual]	CBT-AN	BMI Eating Disorder: EDE-Q Behavior to Physical Exercise: EBQ; CET; CES Depression and anxiety: K-10 Motivation to change: ANSOCQ QOL: EDQOL; HRQOL-12	↑ BMI (16.58 ± 1.04 vs. 16.47 ± 1.2 at baseline; to 16.99 ± 4.04 vs. 18.49 ± 2.9 at 34 wk; *p* = 0.01) ≠EDE (from 3.54 ± 1.29 vs. 3.18 ± 1.29 at baseline; to 2.28 ± 1.32 vs. 2.16 ± 1.42 at 34 wk) ≠EBQ (from 45.6 ± 21.2 vs. 46.0 ± 24.3 at baseline; to 27.1 ± 21.1 vs. 40.0 ± 27.5 at 34 wk) ↑ CET (from 15.7 ± 4.3 vs. 16.8 ± 4.5 at baseline; to 11.9 ± 5.5 vs. 14.1 ± 4.8 at 34 wk; *p* < 0.06) ≠CES (from 60.8 ± 27.9 vs. 70.1 ± 26.7 at baseline; to 36.8 ± 26.5 vs. 47.8 ± 30.4 at 34 wk) ≠K-10 (from 31.5 ± 9.2 vs. 30.4 ± 9.9 at baseline; to 24.3 ± 8.4 vs. 22.3 ± 9.8 at 34 wk) ≠ANSOCQ (from 2.4 ± 0.6 vs. 2.4 ± 0.6 at baseline; to 3.0 ± 0.9 vs. 3.3 ± 1.0 at 34 wk) ↑ EDQOL (from 1.6 ± 0.6 vs. 1.8 ± 0.8 at baseline; to 1.1 ± 0.8 vs. 1.0 ± 1.0 at 34 wk; *p* < 0.05) ≠HRQoL (from 28.7 ± 10.9 vs. 29.5 ± 13.0 at baseline; to 35.6 ± 10.0 vs. 39.1 ± 11.4 at 34 wk)	Not reported

AMA: Arm Muscle Area; AN: Anorexia Nervosa; ANSOCQ: Motivation to change 20-item AN Stages of Change Questionnaire; BAT: Body Attitude Test; BF:% Body Fat; BN: Bulimia Nervosa; BW: Body Weight; CBT: Cognitive Behavioural Therapy; CES: Commitment to Exercise Scale; CES-D: Epidemiological Studies Depression Scale; CET: Compulsive Exercise Test; CG: Control Group; EAT-40: Eating Attitude Test-40; EBQ: Exercise Beliefs Questionnaire; EDE-Q: Eating Disorder Examination Questionnaire; EDNOS: Eating Disorders Not Otherwise Specified; EDI: Eating Disorders Inventory; EDQOL: Eating Disorder Quality of Life; EG: Experimental Group; HRQOL-12: Health-Related Quality of Life (Short Form-12); BMI: Body mass index; K-10: Depression and anxiety symptoms, Kessler-10 item scale; min: Minute; MTMA: Mid-Thigh Muscle Area; N: Number of participants; n: Sample group size; p: Probability value; POMS: Profile of Mood States; SMM: Skeletal Muscle Mass; SF-36: Medical Outcomes Survey Short-form QoL questionnaire; STAI: State Trait Anxiety Inventory; 6RM: 6 Maximum Repetition; wk: Week. ↑: Improvement EG vs. CG; ↓: Worse EG vs. CG; ≠: No differences EG vs. CG; =: No changes EG.

**Table 2 ijerph-19-13921-t002:** Risk of bias. Cochrane Collaboration’s tool for assessing risk of bias [28].

Study	Sequence	Allocation	Blinding1	Blinding2	Outcome1	Outcome2	Other
Thien V et al., 2000 [30]	(+)	(+)	(-)	(-)	(+)	(+)	(+)
Hart S et al., 2001 [31]	(+)	(+)	(-)	(-)	(-)	(-)	(+)
Del Valle MF et al., 2010 [32]	(+)	(+)	(-)	(-)	(+)	(+)	(+)
Catalan-Matamoros D et al., 2011 [33]	(+)	(+)	(-)	(-)	(+)	(+)	(+)
Fernandez-Del-Valle M et al., 2014; 2015; 2016 [34,35,36]	(+)	(-)	(-)	(-)	(+)	(+)	(+)
Hay P et al., 2018 [37]	(+)	(+)	(+)	(+)	(+)	(+)	(-)

Sequence: random sequence generation; Allocation: allocation concealment; Blinding1: blinding of participants and personnel; Blinding2: blinding of outcome assessment; Outcome1: incomplete outcome data; Outcome2: selective reporting. Other: Design issues or initial imbalance. Low risk (+); High risk (-).

**Table 3 ijerph-19-13921-t003:** Quality of evidence GRADE of included studies [29].

No. of Studies	Study Design	Risk of Bias	Inconsistency	Indirect Evidence *	Inaccuracy	Publication Bias **	Impact	Quality	Importance ***
Physiological and muscular effects (measured with BMI) (Follow-up 8 weeks–8.5 months)									
4 [30,32,34,35,36,37]	randomized trials	serious ^a^	serious ^b^	no serious	serious ^c^	no serious	serious ^c^	⊕⊕⊕⊝ MODERATE	4/9
Physiological and muscular effects (measured with muscle strength) (Follow-up 8–12 weeks)									
2 [32,34,35,36]	randomized trials	serious ^a^	no serious ^d^	no serious	no serious ^d^	no serious	no serious ^d^	⊕⊕⊕⊝ MODERATE	5/9
Physiological and muscular effects (measured with BF) (Follow-up 8–12 weeks)									
3 [30,32,34,35,36]	randomized trials	serious ^a^	serious ^e^	no serious	serious ^e^	no serious	serious ^e^	⊕⊕⊝⊝ LOW	2/9
Physiological and muscular effects (measured with AMA and MTMA) (Follow-up 8 weeks)									
1 [34,35,36]	randomized trial	serious ^a^	no serious ^f^	no serious	no serious ^f^	no serious	no serious ^f^	⊕⊕⊝⊝ LOW	3/9
Psychological effects (depression and anxiety measured with K-10 and STAI) (Follow-up 5 weeks–8.5 months)									
2 [31,37]	randomized trials	serious ^a^	serious ^g^	no serious	serious ^g^	no serious	no serious ^g^	⊕⊕⊝⊝ LOW	3/9
Cognitive effects (mood measured with POMS) (Follow-up 5 weeks)									
1 [31]	randomized trial	serious ^a^	no serious ^h^	no serious	no serious ^h^	no serious	no serious ^h^	⊕⊕⊕⊝ MODERATE	4/9
Cognitive effects (attitude measured with BAT and EAT-40) (Follow-up 7 weeks)									
1 [33]	randomized trial	serious ^a^	no serious ^i^	no serious	no serious ^i^	no serious	no serious ^i^	⊕⊕⊕⊝ MODERATE	4/9
Behavioral effects (eating measured with EDI, ANSOQ and EDE-Q) (Follow-up 5 weeks–8.5 months)									
3 [31,33,37]	randomized trials	serious ^a^	no serious ^j^	no serious	no serious ^j^	no serious	no serious ^j^	⊕⊕⊕⊝ MODERATE	4/9
Behavioral effects (impulse control measured with CET) (Follow-up 8.5 months)									
1 [37]	randomized trial	no serious	no serious ^k^	no serious	no serious ^k^	no serious	no serious ^k^	⊕⊕⊝⊝ LOW	2/9
Quality of life effects (measured with SF-36, EDQOL and HRQOL-12) (Follow-up 12 weeks–8.5 months)									
2 [30,37]	randomized trials	serious ^a^	no serious ^l^	no serious	no serious ^l^	no serious	no serious ^l^	⊕⊕⊕⊝ MODERATE	4/9

AMA: Arm Muscle Area; ANSOQ: Motivation to change 20-item AN Stages of Change Questionnaire; BAT: Body Attitude Test; BF: Body fat; BMI: Body Mass Index; CET: Compulsive Exercise Test; EAT-40: Eating Attitude Test-40; EDE-Q: Eating Disorder Examination Questionnaire; EDI: Eating Disorders Inventory; EDQOL: Eating Disorder Quality of Life; EG: Experimental Group; HRQOL-12: Health-Related Quality of Life (Short Form-12); K-10: Depression and anxiety symptoms Kessler-10 item scale; MTMA: Midthigh muscle area; POMS: Profile of Mood States; SF-36: Medical Outcomes Survey Short-form QoL questionnaire; STAI: State Trait Anxiety Inventory. * Considered no serious because, all patients were well diagnosed, and at least 70% with anorexia nervosa. ** Considered no serious because, the study is published in an indexed international journal, and the article reports significant and not significant results. *** Importance: Range from 1—not important, to 9 it is considered critical. (Importance range: from 7 to 9: critical; from 4 to 6: important; from 1 to 3: limited importance). ^a^. Although the studies had a suitable randomly sequence, blinding of assessors is only fulfilled in one of the studies or it is not fulfilled. ^b^. Most of studies (3 of 4 trials) showed no benefits of the treatment vs control group. ^c^. Only one of the 4 studies report significant changes in BMI. ^d^. Muscle strength show increase in both studies. ^e^. BF does not improve in any of the three studies. ^f^. Both the AMA and MTMA show benefits in the study. ^g^. Only in one study decreases anxiety in experimental group, whereas in the shorter duration (5 weeks). ^h^. Mood shows benefits in experimental group in the study. ^i^. Attitudes show benefits in experimental group in the study. ^j^. Eating attitude improves in two of the three studies. ^k^. Control compulsive exercise shows benefits in experimental group in the study. ^l^. Quality of life improves in experimental group in both studies.

## Data Availability

Not applicable.

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
