# Peer review of "Physical Therapy Interventions in Patients with Anorexia Nervosa: A Systematic Review"

_ijerph, 2022, doi:10.3390/ijerph192113921_

Round 1

Reviewer 1 Report

Manuscript Physical therapy interventions in patients with anorexia nervosa: A systematic review

The paper is a review of the very few published works on the effectiveness and usefulness of physical therapies in AN, in the light of the preponderant symptom of physical hyperactivity, which is present in the vast majority of female patients (more than 80%).

In the Introduction, line 40: it may be misleading to describe AN as a body dysmorphic disorder. I would rather say that it is characterized by a disturbance of body image (i.e., affected individuals overestimate their body weight and shape, despite severe underweight).

If on the one hand the physical interventions mentioned in the introduction, from line 70 to line 73, can reduce eating disorders symptoms (some bibliographic references would be needed), on the other hand they require careful evaluation, as they must necessarily be accompanied by an increase in daily caloric intake, otherwise weight loss and thus clinical worsening will occur.

The “years of treatment” are never reported in column two (table 1), perhaps the entry should be deleted.

It is not clear whether the physical sessions were carried out in group or individual format.

Why is the BMI higher in the treatment groups than in controls (18.1 vs 17.59)? 

Considering AN diagnosis, BMIs reported appear particularly high. Was it the authors' choice to propose such treatment to patients who were in a weight range a little below average?

The reference in the text to the tables (e.g., 2 and 3) in the supplementary material should be specified.

Some key characteristics of the samples are missing or are not emphasized (age of onset, comorbidity, illness duration, context: out-patients, in-patients, DH, …).

A more detailed description of the interventions is missing, and I would underline the fact that physical interventions considered in the present review are very heterogeneous … please, discuss this point, because it can reduce generalizability of results. 

The topic is interesting, but it would be useful to give some more clinical background about when to use this type of treatment, with which patients and what to expect.

Author Response

Dear Reviewer,

We appreciate the time and effort put into commenting on the revision of our manuscript, which has resulted in a much stronger manuscript. We agree with the suggested corrections and the changes have been made.

All editions are written in the text using blue letters. 

We hope that the revised manuscript is now suitable for publication in IJERPH 

Yours sincerely,

The authors.

Manuscript Physical therapy interventions in patients with anorexia nervosa: A systematic review.

The paper is a review of the very few published works on the effectiveness and usefulness of physical therapies in AN, in the light of the preponderant symptom of physical hyperactivity, which is present in the vast majority of female patients (more than 80%).

Response: We sincerely appreciate all the valuable comments and suggestions.

Please see below our responses to each of your comments.

In the Introduction, line 40: it may be misleading to describe AN as a body dysmorphic disorder. I would rather say that it is characterized by a disturbance of body image (i.e., affected individuals overestimate their body weight and shape, despite severe underweight).

Response:  We agree with the proposal, and we have modified the text on page 1:

AN is characterized by an active fight against the feeling of hunger, along with disturbance of body image and an intense fear of weight gain or obesity, even when their weight is clearly low.

If on the one hand the physical interventions mentioned in the introduction, from line 70 to line 73, can reduce eating disorders symptoms (some bibliographic references would be needed), on the other hand they require careful evaluation, as they must necessarily be accompanied by an increase in daily caloric intake, otherwise weight loss and thus clinical worsening will occur.

Response: We agree with the proposal, and we have we have modified text and citations on page 2:

As supervised physical therapy might increase weight and improve the overall condition in AN patients through therapeutic aerobic exercising, massage, basic body awareness therapy and yoga, these interventions could reduce the eating disorder in patients with AN and BN (Vancampfort D et al., 2014).

The “years of treatment” are never reported in column two (table 1), perhaps the entry should be deleted.

Response: Mean age (years treatment/control), it is the mean age of the participants in experimental group and control group. To make it clearer, we have added the heading of the column two in table 1:

Mean age (years EG/CG)

It is not clear whether the physical sessions were carried out in group or individual format.

Response: Thien et al. and Hart et al. they do not clarify whether the sessions were individual or group. Catalan et al. they performed a first individual session, and the rest in a group. The other studies, all the sessions were individualized.

To inform about this aspect, we have added the following sentence in the heading of column 4:

[Training: individual, group, or unspecified]

And in each box, it has been specified how the exercise treatment sessions were performed.

Why is the BMI higher in the treatment groups than in controls (18.1 vs 17.59)?

Response: These data come from the primary studies included in our Systematic Review. It is the mean BMI according to the experimental and control groups of the studies.

Considering AN diagnosis, BMIs reported appear particularly high. Was it the authors' choice to propose such treatment to patients who were in a weight range a little below average?

Response: Thank you for pointing this out. For clarification, before the paragraph beginning with: “Across the six eligible studies, from out of the eight articles, patients with AN had a mean BMI of 18.10 (experimental group) and 17.59 (control group)”; We have introduced the following text on line 52 of page 11:

In relation to the BMI of the patients in the primary studies included in our SR, Thien et al. (Thien V et al., 2000), Catalan-Matamoros et al. (Catalan-Matamoros D et al., 2011), and Hart et al. (Hart S et al., 2001) they do not specify a BMI in their inclusion criteria. In terms of the studies of Del Valle et al. (del Valle MF et al., 2010; Fernandez-Del-Valle M et al., 2014; Fernandez-del-Valle M et al., 2015; Fernández-Del-Valle M et al., 2016) and they re-quired a BMI higher than 14, and the study of Hay et al. (Hay P et al., 2018) the patients could have a BMI range between 14 and 18.5; and therefore, the thinness would be be-tween mild and severe according to WHO.

The reference in the text to the tables (e.g., 2 and 3) in the supplementary material should be specified.

Response: Thank you very much for the comment, we have added the references in the text to the table 2 and table 3, which have been incorporated into the main text, since, by mistake on our part, they had been included as supplementary material.

Some key characteristics of the samples are missing or are not emphasized (age of onset, comorbidity, illness duration, context: out-patients, in-patients, DH, …).

Response: We agree with the reviewer that it would be very interesting to be able to analyse or compare the comorbidities or the duration of the disease; However, this information is not always available in the studies. When we have performed the analysis of each of the clinical trials, we have extracted the information that was available in them.

Regarding the hospitalization of patients, in the text on line 31, page 12, it states the following:

90.82% of the patients (188) were not hospitalised (Catalan-Matamoros D et al., 2011; del Valle MF et al., 2010; Fernandez-Del-Valle M et al., 2014; Fernandez-del-Valle M et al., 2015; Fernández-Del-Valle M et al., 2016; Hart S et al., 2001; Hay P et al., 2018; Thien V et al., 2000); and one study with 19 patients included hospitalised and non-hospitalised patients, although the number of each is not specified (Hart S et al., 2001).

A more detailed description of the interventions is missing, and I would underline the fact that physical interventions considered in the present review are very heterogeneous … please, discuss this point, because it can reduce generalizability of results.

Response: All primary studies include treatment with physical therapy modalities. Indeed, there are heterogeneous interventions in terms of content and duration, and we have reflected this as a limitation (page 19). However, we consider it noteworthy that we have been able to show that thanks to the Systematic Review performed.

The topic is interesting, but it would be useful to give some more clinical background about when to use this type of treatment, with which patients and what to expect.

Response: Thank you for pointing this out. We have tried to point this out, at the beginning of the section Implications for rehabilitation, on page 19 of our manuscript:

AN is a serious and complex psychiatric disorder, with severe somatic consequences, poor prognostic, increase of mortality, decrease quality of life, and in the most serious cases, reducing autonomy and physical capacity.

Supervised physical therapy might increase weight or BMI, and improve muscle strength and cardiovascular endurance, and the overall condition in AN women. In relation to quality of life, there is still controversy as previous SR were not conclusive on this subject.

In previous systematic reviews carried out about eating disorders, less than 40% of participants had diagnosed of AN. This is the only systematic review carried out with more than 85% of participants diagnosed of AN.

Reviewer 2 Report

The article is a systematic review of the effectiveness of physical therapies in the treatment of anorexia nervosa. The article is well written, clear and the analysis of the articles well done. 

I am however surprised not to find some articles in the selection, in particular no article on the effectiveness of yoga which has been well studied. Ad well as pilate : Feasibility and effect of a Pilates program on the clinical, physical and sleep parameters of adolescents with anorexia nervosa.

Martínez-Sánchez SM, Martínez-García TE, Bueno-Antequera J, Munguía-Izquierdo D.Complement Ther Clin Pract. 2020 May;39:101161. doi: 10.1016/j.ctcp.2020.101161. Epub 2020 Apr 2.

And other very good articles on this field : Development and evaluation of an adapted physical activity program in anorexia nervosa inpatients: A pilot study.

Kern L, Morvan Y, Mattar L, Molina E, Tailhardat L, Peguet A, De Tournemire R, Hirot F, Rizk M, Godart N, Fautrelle L.Eur Eat Disord Rev. 2020 Nov;28(6):687-700. doi: 10.1002/erv.2779. Epub 2020 Sep 24.

The search criteria may have been too restrictive.

The study goals should then be revised to include only certain types of physical therapies. But then the interest of the study is strongly diminished.

Author Response

Dear Reviewer,

We appreciate the time and effort put into commenting on the revision of our manuscript, which has resulted in a much stronger manuscript. We agree with the suggested corrections and the changes have been made.

All editions are written in the text using blue letters.

We hope that the revised manuscript is now suitable for publication in IJERPH. 

Yours sincerely,

The authors.

The article is a systematic review of the effectiveness of physical therapies in the treatment of anorexia nervosa. The article is well written, clear and the analysis of the articles well done.

Response: Many thanks for these comments. We highly appreciate your valuable feedback on our manuscript.

Please see below our responses to each of your comments.

I am however surprised not to find some articles in the selection, in particular no article on the effectiveness of yoga which has been well studied. Ad well as Pilates: Feasibility and effect of a Pilates program on the clinical, physical and sleep parameters of adolescents with anorexia nervosa.

And other very good articles on this field: Development and evaluation of an adapted physical activity program in anorexia nervosa inpatients: A pilot study.

Response: We agree with you that they are indeed two very interesting studies. In both cases, these are pre-test/post-test designs with no control group, reason why we could not include them in our Systematic Review, in which we have only included randomized and non-randomized controlled clinical trials designs.

The search criteria may have been too restrictive.

Response: In our Systematic Review we have included randomized clinical trials, study design that is considered gold standard, and controlled clinical trials, because it is not always possible to randomly assign participants to experimental and comparison conditions. Both types of studies are considered to be of higher methodological quality and lower risk of bias. 

All primary studies that have met this quality requirement in their designs have been included in our Systematic Review.

The study goals should then be revised to include only certain types of physical therapies. But then the interest of the study is strongly diminished.

Response: The objectives of our systematic review were to provide an updated of the evidence to effectiveness physiotherapy modalities in anorexia nervosa patients, and to verify the safety of these treatments.

The six studies have been analysed jointly because they are, in all cases, physiotherapy treatments, regardless of the type, dose, or intensity of treatment of the primary studies. 

Round 2

Reviewer 2 Report

The authors have responded point by point to the remarks made. The fact that the search criteria are very restrictives makes the study less interesting and reports few type of physical therapy. Because of this, I don't find that this study brings any advances to the literature. Other previous systematic reviews with broader criteria were, in my opinion more relevant (Davy Vancampfort 2014), although recent articles have been included in this study. The study, however, meets the criteria that the authors set for themselves. 

Author Response

Response: The authors appreciate your opinions. And we agree with the reviewer on the relevance of the study by Vancampfort et al. in 2014, highlighting that it was the first Systematic Review on physical therapy interventions for patients with anorexia and bulimia nervosa.